# Generation of *Slco1a4-Cre^ERT2^*-tdTomato Knock-in Mice for Specific Cerebrovascular Endothelial Cell Targeting

**DOI:** 10.3390/ijms25094666

**Published:** 2024-04-25

**Authors:** Chengfang Xu, Shounian Li, Yunting Cai, Jinjin Lu, Yan Teng, Xiao Yang, Jun Wang

**Affiliations:** 1Beijing Institute of Lifeomics, Beijing 102206, China; xcfjason@163.com (C.X.); yuntingcai2024@126.com (Y.C.); tengyan@bmi.ac.cn (Y.T.); 2College of Life Science, Liaoning University, Shenyang 110036, China; 3Department of Basic Medical Sciences, School of Medicine, Tsinghua University, Beijing 100084, China

**Keywords:** Cre, *Slco1a4*, blood–brain barrier, endothelial cells, lineage tracing

## Abstract

The cerebrovascular endothelial cells with distinct characteristics line cerebrovascular blood vessels and are the fundamental structure of the blood–brain barrier, which is important for the development and homeostatic maintenance of the central nervous system. Cre-LoxP system-based spatial gene manipulation in mice is critical for investigating the physiological functions of key factors or signaling pathways in cerebrovascular endothelial cells. However, there is a lack of Cre recombinase mouse lines that specifically target cerebrovascular endothelial cells. Here, using a publicly available single-cell RNAseq database, we screened the solute carrier organic anion transporter family member 1a4 (*Slco1a4*) as a candidate marker of cerebrovascular endothelial cells. Then, we generated an inducible Cre mouse line in which a CreERT2-T2A-tdTomato cassette was placed after the initiation codon ATG of the *Slco1a4* locus. We found that tdTomato, which can indicate the endogenous *Slco1a4* expression, was expressed in almost all cerebrovascular endothelial cells but not in any other non-endothelial cell types in the brain, including neurons, astrocytes, oligodendrocytes, pericytes, smooth muscle cells, and microglial cells, as well as in other organs. Consistently, when crossing the *ROSA26^LSL-EYFP^* Cre reporter mouse, EYFP also specifically labeled almost all cerebrovascular endothelial cells upon tamoxifen induction. Overall, we generated a new inducible Cre line that specifically targets cerebrovascular endothelial cells.

## 1. Introduction

The blood–brain barrier (BBB) serves as a physical and metabolic interface, allowing for the passage of essential nutrients and preventing the entrance of potentially harmful substances [1,2]. The BBB consists mainly of endothelial cells (ECs) lining the blood vessels, pericytes, and astrocytes. Cerebrovascular ECs, compared to ECs in other tissues, are characterized by specialized tight junctions, low rates of transcytosis, and abundantly expressed specific transporters [3,4]. Subsequently, the unique functions of cerebrovascular ECs were verified by many gene manipulation mouse models based on the Cre-LoxP recombination system. Although several Cre mouse lines were generated to target ECs, such as *Tie2-Cre* and *SP-A-Cre*, only a few can specifically label cerebrovascular ECs [5,6,7]. For instance, we previously generated a transgenic mouse line in which Cre expression was driven by surfactant protein A (SP-A), a marker of alveolar type II cells of the lung. Unexpectedly, Cre activity was also observed in cerebrovascular ECs and gastric epithelium [8]. Furthermore, mice with *SP-A-Cre* mediated *Smad4* deletion in cerebrovascular ECs caused perinatal intracranial hemorrhage (ICH) and death [9]. Therefore, there is a lack of inducible and more specific Cre mouse targeting in cerebrovascular ECs.

Here, to select candidate markers of cerebrovascular ECs, we screened several publicly available single-cell RNA-seq (scRNAseq) databases [10,11] and particularly focused on transporters. Then, *Slco1a4*, a membrane transporter, stood out. *Slco1a4*, also known as *Oatp1a4*, is a member of the solute carrier organic anion transporter family [12]. Several studies have revealed that *Slco1a4* is involved in the hepatic uptake of a wide range of substrate drugs [13,14]. Furthermore, *Slco1a4* is also expressed in brain microvessels and is responsible for the blood-to-brain transport of various drugs [15,16]. Next, we generated an inducible *Slco1a4-Cre^ERT2-T2A-tdTomato^* mouse line, where the *Cre^ERT2-T2A-tdTomato^* expression was driven by the endogenous *Slco1a4* locus. Finally, we found that the endogenous *Slco1a4* expression shown by *tdTomato* expression was restricted in cerebrovascular ECs, and as expected, Cre activity specifically targeted brain ECs.

## 2. Results

### 2.1. Slco1a4 Is a Candidate Marker of Cerebrovascular ECs

To identify novel markers of brain endothelial cells (ECs), we comprehensively screened publicly available scRNAseq databases. First, in the Tabula Muris dataset, which encompasses a single-cell atlas of 20 mouse organs representing 38 distinct cell types, *Slco1a4* expression was only observed in the brain, but not in other organs [11] (Figure 1A). Further analysis revealed that, in the brain, *Slco1a4* exhibited specific expression in ECs, but not other non-EC cell types, including pericytes (PC), smooth muscle cells (SMC), microglia (MG), fibroblast-like cells (FB), and oligodendrocytes (OL) [10] (Figure 1B). Then, the analysis of the endothelial cells scRNAseq atlas derived from the brain revealed that *Slco1a4* was expressed in all brain EC subsets, including arterial, venous, and capillaries ECs [10]. Therefore, Slco1a4 is a potential marker for brain endothelial cells.

### 2.2. Generation of the Slco1a4-Cre^ERT2^-tdTomato Knock-in Mouse Line

We used the CRISPR/Cas9 system to integrate the CreERT2 cassette into the frame with the translation start codon ATG of the endogenous *Slco1a4*, thus replacing the *Slco1a4* expression (Figure 2A). The targeting vector included CreERT2-T2A-tdTomato-rBG pA, along with 5′ and 3′ homologous arms. The T2A self-cleaving peptide enabled the separate production of CreERT2 and tdTomato proteins. Moreover, the presence of tdTomato expression can indicate the endogenous expression profile of *Slco1a4*. The *Slco1a4-Cre^ERT2^* donor vector, guide RNA, and Cas9 mRNA were simultaneously microinjected into fertilized mouse eggs to produce targeted offspring. Southern blot using a probe against the transgenic sequence confirmed the accurate integration of the donor CreERT2 DNA into the *Slco1a4* locus (Figure 2B). Primer pairs 1 and 2 were utilized for distinguishing between wildtype and transgenic genotypes, respectively (Figure 2C).

### 2.3. Cerebrovascular ECs-Specific Expression of Slco1a4

Next, to verify the expression pattern of *Slco1a4*, we detected the *tdTomato* expression by direct fluorescent imaging or immunofluorescence across various tissues of *Slco1a4-Cre^ERT2-T2A-tdTomato^* at postnatal day 56 (P56). First, in the brain, tdTomato signals formed a dense vascular-like network (Figure 3). When co-immunostained with endothelial cells marker CD31, all *tdTomato*-expressing cells were positive for CD31 throughout all of the brain zonation, including the cortex, hippocampus, cerebellum, and thalamus (Figure 3). More importantly, all CD31^+^ ECs were positive for tdTomato, suggesting that *Slco1a4* could specifically label almost all ECs, including arterial, venous, and capillaries ECs (Figure 3). The specific expression of *Slco1a4* in brain ECs was further confirmed, as tdTomato^+^ cells did not express any markers of other cell types, including PDGFRβ^+^ pericytes, α-SMA^+^ smooth muscle cells, NeuN^+^ neurons, GFAP^+^ astrocytes, NG2^+^ oligodendrocytes, and IBA1^+^ microglial cells (Figure 4). Several studies have shown that *Slco1a4* is also expressed in hepatocytes. However, we did not find any tdTomato signal in the liver, bladder, stomach, heart, intestine, eye, or lung (Figure 5A,B). These data suggested that the *Slco1a4* expression is restricted in cerebrovascular ECs.

### 2.4. Cerebrovascular Endothelial Cells-Specific Cre Activity of Slco1a4-Cre^ERT2^ Mice

To examine the Cre activity, we crossed *Slco1a4-Cre^ERT2^* mice with the *ROSA26^LSL-EYFP^* (short for ROSA26-loxP-stop-loxP-EYFP) Cre reporter mice (Figure 6A). In *Slco1a4-Cre^ERT2^* and *ROSA26^LSL-EYFP^* mice, Cre recombinase was fused to the mutant ligand-binding domain of the estrogen receptor (ERT2), which only binds to tamoxifen (TAM), a selective estrogen receptor modulator, but not the endogenous estrogen [17]. Without TAM, the CreERT2 fusion protein was restricted to the cytoplasm, while upon TAM administration, TAM drives CreERT2 to translocate into the nucleus [18,19]. Subsequently, Cre recombinase can delete the STOP sequence flanked by two loxP sites, leading to the constitutive expression of *EYFP* at the *ROSA26* locus. To activate Cre activity, *Slco1a4-Cre^ERT2^* and *ROSA26^LSL-EYFP^* mice at P42 were administered TAM over 8 consecutive days and harvested at P56 (Figure 6B). As expected, we found that all EYFP^+^ cells in the brain were positive for tdTomato or CD31, confirming the specific expression of *Slco1a4-Cre^ERT2^* activity in brain ECs (Figure 6C). Of note, all tdTomato^+^ or CD31^+^ cells were labeled by EYFP, indicating the sufficient induction of Cre activity with this strategy. Finally, we did not find tdTomato^+^ cells in the absence of TAM treatment, indicating no leakiness of Cre activity in *Slco1a4-Cre^ERT2^* and *ROSA26^LSL-EYFP^* mice (Figure 7).

## 3. Discussion

Here, we generated a new inducible brain ECs-specific Cre mouse line. Taking advantage of several scRNAseq databases, we selected a transporter Slco1a4 as a candidate marker of brain ECs [10,11]. In vivo results confirmed that *Slco1a4* expression exhibited specificity in brain ECs not only compared with other tissue ECs but also compared with other cell types in the brain. More importantly, besides the specificity, *Slco1a4* was expressed in almost all brain ECs throughout whole brain regions, suggesting that a single gene, *Slco1a4*, can specifically and efficiently label brain ECs. Therefore, the *Slco1a4*-driven Cre mouse line would be a new powerful tool for precise gene manipulation in brain ECs.

Numerous mouse lines have been established to target ECs, such as *Tie2-Cre* and *SP-A-Cre*. Although mice with *Tie2-Cre*-mediated *PTEN* deletion in all ECs died before embryonic day 11.5 [20], mice with *SP-A-Cre*-mediated *PTEN* deletion in brain ECs can survive to adulthood [21]. Then, taking advantage of this *SP-A-Cre*, we demonstrated that *PTEN* in brain ECs upregulates *MCT1* to enhance lactate transport across the brain endothelium, which is critical for lactate homeostasis, adult hippocampal neurogenesis, and cognitive function [21]. Unfortunately, *SP-A-Cre* can also induce *PTEN* deletion in the gastric epithelium, which could lead to gastric tumors, progressive weakness, and a loss in body weight after the age of 1 month, preventing the further investigation of the *PTEN* function in aged mouse brain ECs. The *Slco1c1-Cre^ERT2^* BAC transgenic mice have also been reported to target cerebrovascular ECs and have been effectively utilized for the targeted deletion of disease genes to establish a brain-specific cerebrovascular phenotype [22,23,24,25]. Unfortunately, *Slco1c1-Cre^ERT2^*-induced recombination was also found in the aorta, heart, testis, and choroid plexus epithelial cells [25]. Therefore, there is a lack of inducible and more specific Cre mouse targeting in cerebrovascular ECs. A previous study has reported an inducible brain ECs-specific Cre mouse line *Mfsd2a-CreER*, in which the CreER sequence was knocked in the *Mfsd2a* locus. Upon TAM treatment, *Mfsd2a-CreER* activity was observed not only in brain ECs but also in many neurons, as well as in a large number of hepatocytes and intestinal epithelial cells [26]. To restrict *Mfsd2a-CreER* within brain endothelial cells but not in other cell types, especially neurons, another study used a new intersectional genetic targeting system by combining the Dre-rox and Cre-loxP systems, in which there is sequential Dre-mediated Cre expression in targeted cells that have expressed both Dre and Cre recombinases [27]. Indeed, the compound *Tie2-Dre* and *Mfsd2a-CrexER* line labeled brain ECs efficiently and specifically, without labeling any neurons [26,27]. Here, although the *Cre* expression was driven by a single gene, *Slco1a4*, the *Slco1a4-Cre^ERT2^* line can also efficiently and specifically target brain ECs. Moreover, Cre is the ultimate readout of the *Tie2-Dre* and *Mfsd2a-CreER* line, and after Dre-rox recombination, Cre was constitutive but not inducible. Here, we showed that TAM induction with 8 days at the adult stage is enough for releasing Cre activity in almost all brain ECs of the *Slco1a4-Cre^ERT2^* line.

While the knock-in method enables Cre recombinase to be inserted directly into the locus of the *Slco1a4* gene, allowing Cre expression to be regulated by all of the endogenous control elements, which promotes physiological levels of *Cre^ERT2^* expression, this approach inevitably disrupts the *Slco1a4* gene itself. Studies utilizing *Slco1a4* knockout mice have demonstrated that the absence of *Slco1a4* compromises the brain-to-blood transport of certain statins and affects the hepatic uptake of various substrates [16,28]. Consequently, to avoid the potential effects that may arise from complete *Slco1a4* disruption, it is essential to employ heterozygous rather than homozygous knock-in mice for gene deletion experiments.

In summary, the generation of *Slco1a4-Cre^ERT2^*-tdTomato mice offers a valuable tool for facilitating in-depth investigations into the intricate genetic mechanisms governing the BBB function and its maintenance, thereby offering insights into the fundamental mechanisms underlying neurological diseases associated with BBB dysfunction.

## 4. Materials and Methods

### 4.1. Generation of Slco1a4-Cre^ERT2-T2A-tdTomato^

The *Slco1a4-Cre^ERT2-T2A-tdTomato^* (*Slco1a4-Cre^ERT2^* mice) were generated via the CRISPR/Cas9 approach on the C57BL/6J background. The mouse *Slco1a4* gene (NCBI Reference Sequence: NM_030687.2) is located on mouse chromosome 6. A total of 17 exons have been identified, with the ATG start codon in exon 4 and the TGA stop codon in exon 17 (Transcript: ENSMUST00000165990). The gRNA (gRNA1: GAAAGAGGTTGCAACCCATGGGG; gRNA2: TAGAGGGTCTTAAAGAATAGTGG) to mouse *Slco1a4* gene, the donor vector containing the “CreERT2-T2A-tdTomato-rBG pA” cassette, and Cas9 mRNA were co-injected into fertilized mouse eggs to generate targeted knock-in offspring. The targeted insertion of the CreERT2-T2A-tdTomato cassette replaced the coding sequence starting from exon 4, ensuring *Cre^ERT2^* expression under the endogenous *Slco1a4* promoter. The targeting vector was designed with homology arms spanning regions flanking exon 4, synthesized from a BAC clone template, to facilitate site-specific integration. F0 pups underwent PCR screening and sequencing to confirm the precise integration of the cassette. Subsequently, founder mice with confirmed germline transmission were bred to produce F1 offspring. Successful integration in offspring was confirmed through genotyping with PCR and Southern blot analysis.

### 4.2. Mouse Strains

Mice carrying the *Slco1a4-Cre^ERT2-T2A-tdTomato^* mouse line and *ROSA26^LSL-EYFP^* reporter mice (RRID: IMSR_JAX:006148) [29] were used for the experiments. Littermates were used in all experiments. All experimental procedures were performed using littermates to ensure genetic consistency. Animal care and treatment were in strict compliance with institutional guidelines.

### 4.3. Experimental Design

The animals used in this study were randomized to ensure an unbiased selection for experimentation. Furthermore, the principles of the 3Rs (Replacement, Reduction, and Refinement) were strictly adhered to to minimize animal suffering and use. The minimum sample size required to yield statistically significant results was utilized for all experiments.

### 4.4. Tamoxifen Administration

Cre recombinase activity in P42 mice was induced with tamoxifen (Sigma-Aldrich, T5648, Saint Louis, MO, USA). We intraperitoneally injected a solution of 10 mg/mL tamoxifen (dissolved in 10% ethanol/corn oil) at a dosage of 10 mg/kg body weight daily for 8 days.

### 4.5. Genotyping by PCR

Mouse genomic DNA was extracted from the mouse tail. Mouse tails were lysed by incubation with proteinase K overnight at 55 °C, followed by centrifugation at 12,000× *g* for 10 min to obtain the supernatant with genomic DNA. Genomic DNA was washed with 100% ethanol and 75% ethanol. Specific primers were used to distinguish the mutant allele from the wild-type allele. Primer 1 (Forward: 5′-AGCACCCTAGCCCTAAGAGA-3′, Reverse: 5′-CACTAGATTGACCTGTGACCTCAT-3′) was used to distinguish the wild-type allele (878 bp). Primer 2 (Forward: 5′-TGAATAAAGCACCCTAGCCCTAAGA-3′; Reverse: 5′-TTATTCAACTTGCACCATGCCG-3′) was used to distinguish the mutant allele (461 bp). PCR was carried out in a 20 μL volume for 33 cycles under standard conditions, with the two primers listed above added to each reaction. PCR was carried out in a 20 μL volume for 33 cycles under standard conditions, with the two primers listed above added to each reaction.

### 4.6. Immunostaining

The mice were euthanized and then transcardially perfused with saline/PBS. The tissues were fixed for 8–10 h in 4% PFA and then equilibrated until they sunk to the bottom in 30% sucrose. Thereafter, the tissues were either embedded in paraffin and sectioned at 6 μm or sectioned at 40 μm in OCT (SAKURA) at −30 °C. The sections were treated with endogenous peroxidase blockers for 20 min at room temperature and then blocked with 10% goat serum for 30 min at 37 °C. The sections were incubated with the following primary antibodies overnight at 4 °C: CD31 (1:100, BD Biosciences, 550274, San Jose, CA, USA), GFP (1:500, Cell Signaling Technology, 2956s, EYFP can be recognized by the GFP antibody, Danvers, MA, USA), RFP (1:500, Rockland, Limerick, PA, USA, 600-401-379; tdTomato can be recognized by the RFP antibody), PDGFRβ (1:400, Abcam, ab69506, Cambridge, MA, USA), α-SMA (1:500, Abcam, ab5694), NeuN (1:500, Cell Signaling Technology, 24307S, Danvers, MA, USA), GFAP (1:600, Cell Signaling Technology, 12389S, Danvers, MA, USA), IBA1 (1:500, Abcam, ab178846, Cambridge, MA, USA), and NG2 (1:400, Abcam, ab259324, Cambridge, MA, USA). Briefly, the sections were incubated with the primary antibody, followed by detection using the HRP-conjugated secondary antibody and Tyramide Signal Amplification (TSA)-fluorophores (Histova Biotechnology, NECC7100); then, they were subjected to confocal microscopy on the LSM 880 microscope (Carl Zeiss AG, Oberkochen, Germany).

### 4.7. Southern Blot

The gene targeting in four F1 animals (15, 19, 20, and 22) was confirmed by the Southern blot analysis of the tail DNA samples. ScaI and EcoRV were used to digest genomics DNAs. After electrophoresis, the DNA on the gel was denatured, and the single-stranded DNA fragments were transferred to nylon membranes or other solid phase supports in situ, fixed by dry baking or ultraviolet irradiation, and then hybridized with the labeled probes of the corresponding structure. The content of specific DNA molecules was detected by autoradiography or enzymatic reaction. KI probe forward primer: 5′-TGCCCTGGCTCACAAATACCACT-3′; KI probe reverse primer: 5′-TAGCCAACCTTTGTTCATGGCAGC-3′.

### 4.8. Statistics

The data of each experiment were repeated more than three times and obtained from three different experimental samples. The data were analyzed using GraphPad Prism 8 software and a two-tailed unpaired Student’s *t*-test. The error bars on the graphs represent the mean ± standard deviation (SD). * *p* < 0.05, ** *p* < 0.01, and *** *p* < 0.001 were considered statistically significant.

## Figures and Tables

**Figure 1 ijms-25-04666-f001:**
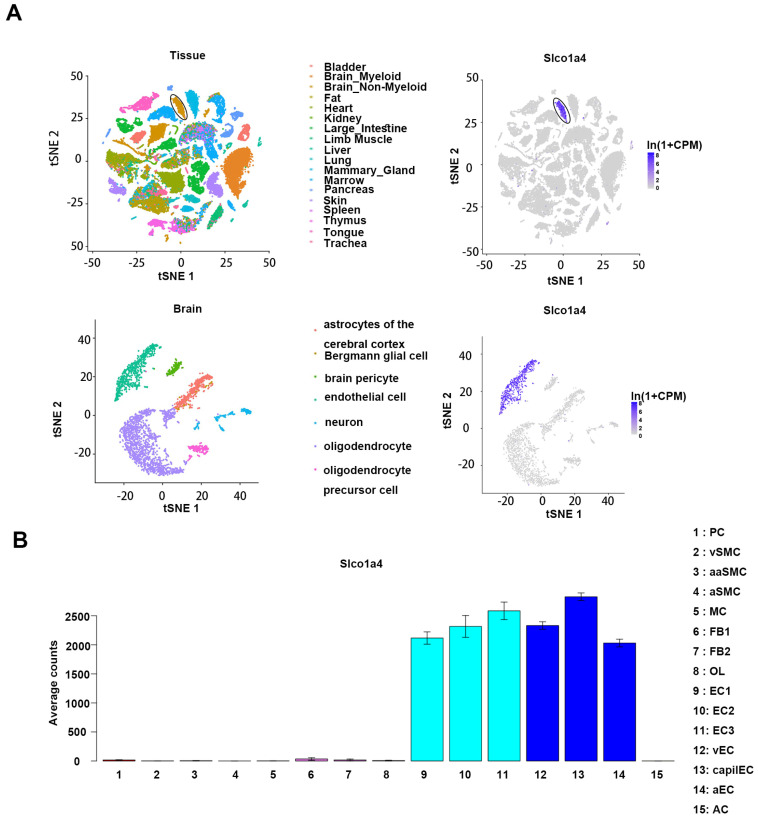
(**A**) Single-cell transcriptomics of 20 mouse organs show *Slco1a4*-specific expression in brain endothelial cells. t-SNE visualization of all FACS cells from 20 tissues (upper) and brain non-myeloid cells (lower). And expression of Slco1a4 (grey/low to purple/high) for each cell type. [11] (**B**) *Slco1a4* expression in adult mouse brain vascular and perivascular cells [10].

**Figure 2 ijms-25-04666-f002:**
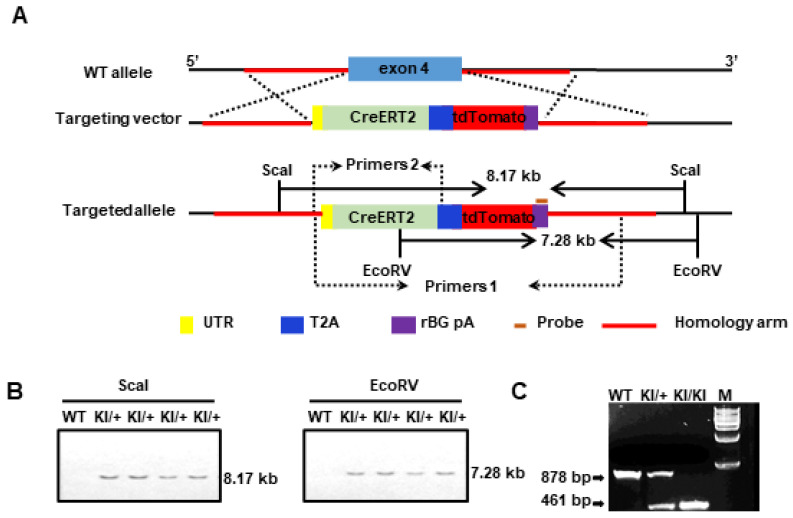
(**A**) Schematic models of the knock-in strategies for generating *Slco1a4-Cre^ERT2^*-tdTomato mouse lines via the CRISPR/Cas9 system; (**B**) Southern blot analysis of the *Slco1a4-Cre^ERT2^*-tdTomato knock-in allele. ScaI-digested genomic DNAs from wild-type (+/+) and heterozygous (KI/+) mice were hybridized with the probe, detecting an 8.17 kb band from the targeted allele. EcoRV-digested genomics DNAs hybridized the probe, detecting a 7.28 kb band from the targeted allele; (**C**) PCR genotyping of *Slco1a4-Cre^ERT2^* wild-type (878 bp) and knock-in (461 bp) mice. Primer 1 was used to distinguish the wild-type allele and Primer 2 was used to distinguish the mutant allele. M is a 1 kb DNA Marker.

**Figure 3 ijms-25-04666-f003:**
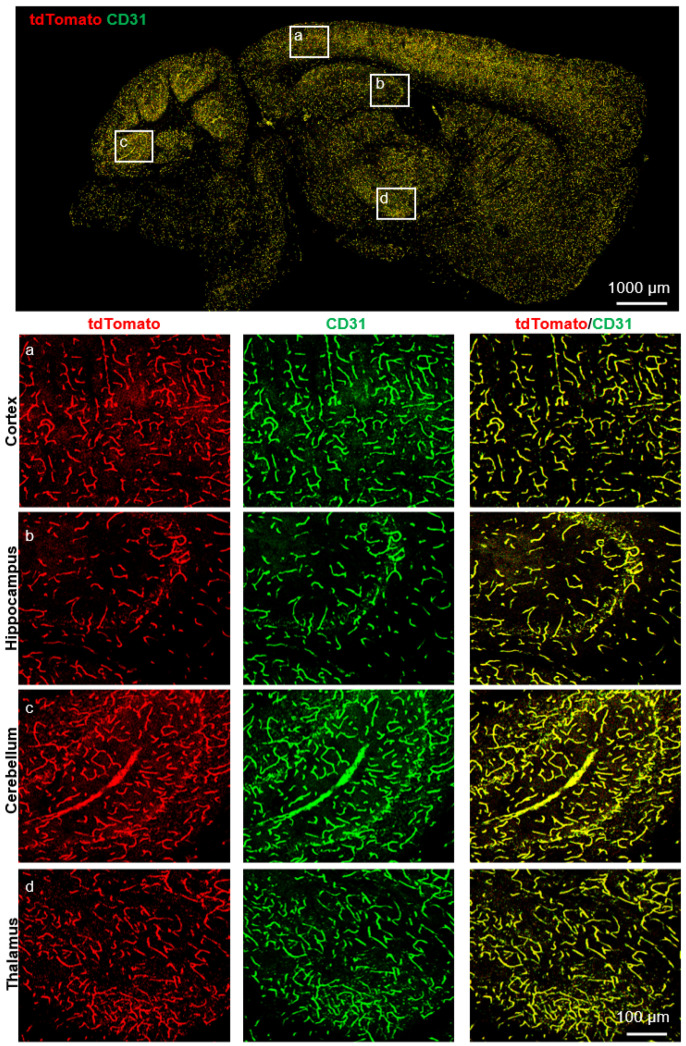
*TdTomato* was expressed in the cerebrovascular ECs of the *Slco1a4-Cre^ERT2^* knock-in mouse line. Immunostaining of brain slices for tdTomato (red) and CD31 (green) in *Slco1a4-Cre^ERT2^* knock-in mice at P56, confirming that tdTomato was detected in the endothelial cells of the cortex, hippocampus, cerebellum, and thalamus at P56. Scale bar: 100 μm.

**Figure 4 ijms-25-04666-f004:**
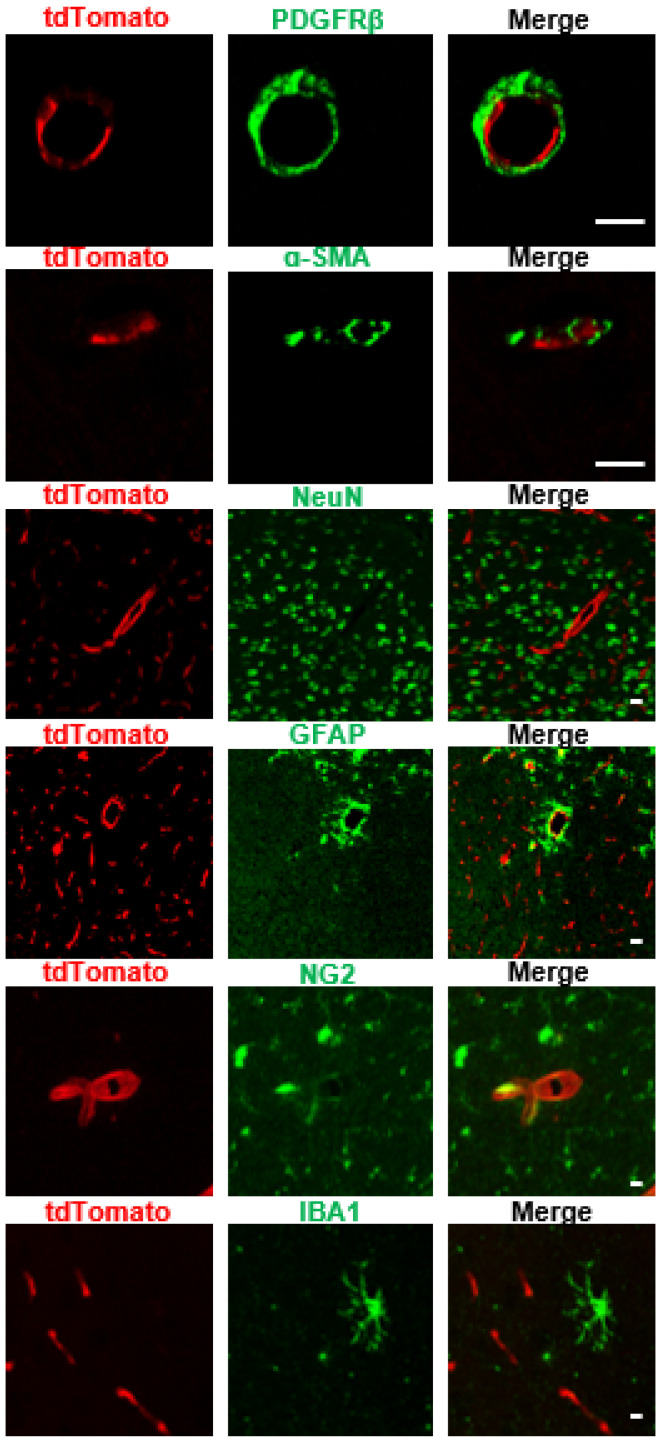
*Tdtomato* was not expressed in any other non-EC cell types in the brain. Immunofluorescence analysis showed that *tdTomato* was not expressed in the pericytes (PDGFRβ^+^), smooth muscle cells (α-SMA^+^), neuron (NeuN^+^), astrocytes (GFAP^+^), oligodendrocytes (NG2^+^), or microglial cells (IBA1^+^). Scale bar = 10 μm.

**Figure 5 ijms-25-04666-f005:**
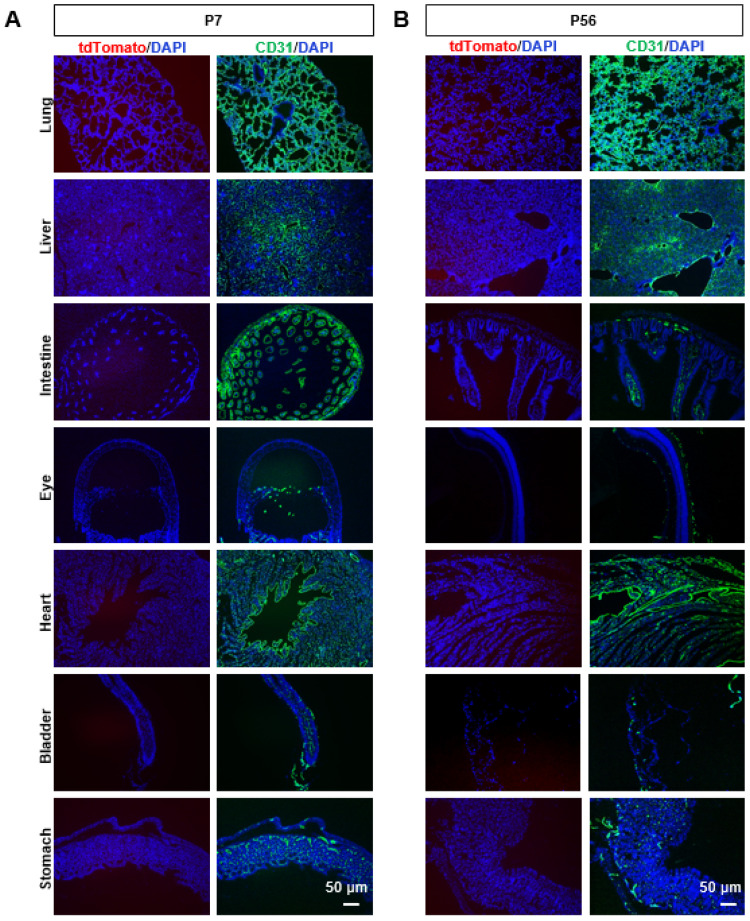
*TdTomato* was not expressed in the lung, liver, gut, eye, heart, bladder, or stomach. (**A**) TdTomato (red) and CD31 (green) immunostaining analysis in the lung, liver, gut, eye, heart, bladder, and stomach at P7. Scale bar = 50 μm. (**B**) TdTomato (red) and CD31 (green) immunostaining analysis in the lung, liver, gut, eye, heart, bladder, and stomach at P56. Scale bar = 50 μm.

**Figure 6 ijms-25-04666-f006:**
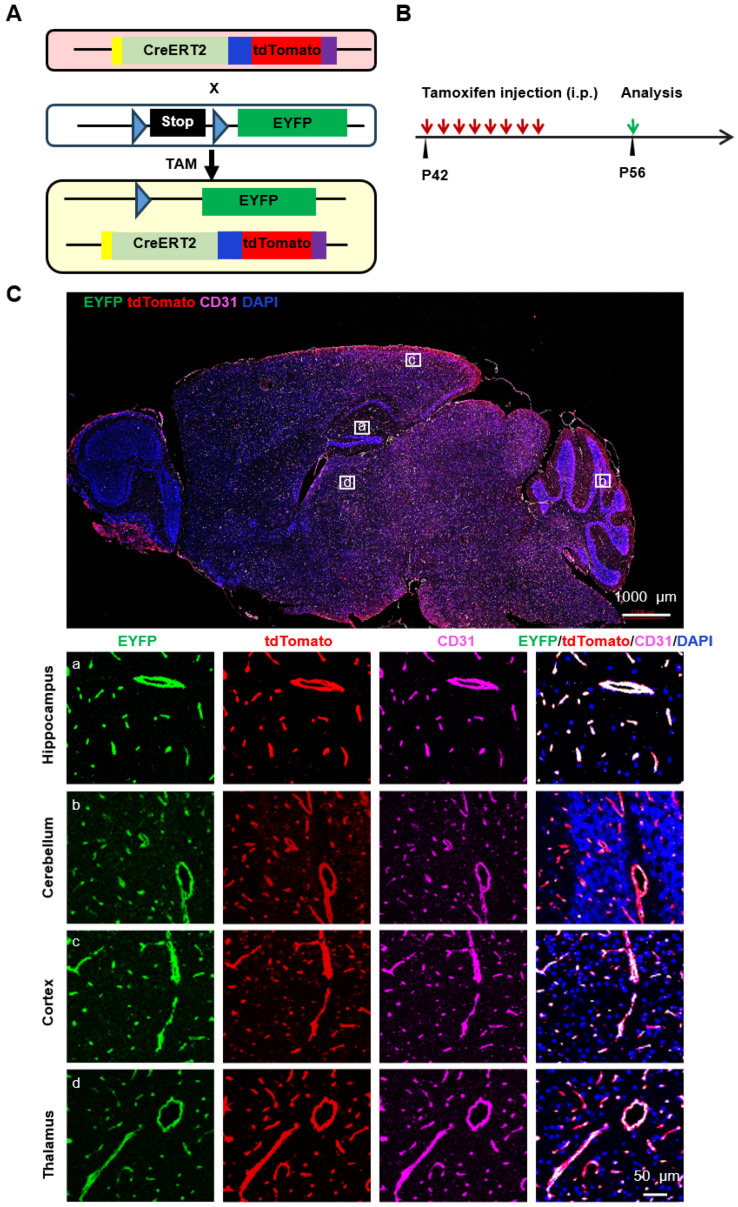
Cre recombinase activity in *Slco1a4-Cre^ERT2^*-tdTomato mice assessed with the *ROSA26^LSL-EYFP^* reporter. (**A**) Schematic showing the crossing of *Slco1a4-Cre^ERT2^* with *ROSA26^LSL-EYFP^* mice. (**B**) Schematic illustration of the experimental protocol. Tamoxifen was administrated at P42 once every day seven times (red arrow), followed by analysis at P56 (green arrow). (**C**) EYFP (green), tdTomato (red), and CD31 (purple) immunostaining analysis in the brain of *Slco1a4-Cre^ERT2^* knock-in mice at P56.

**Figure 7 ijms-25-04666-f007:**
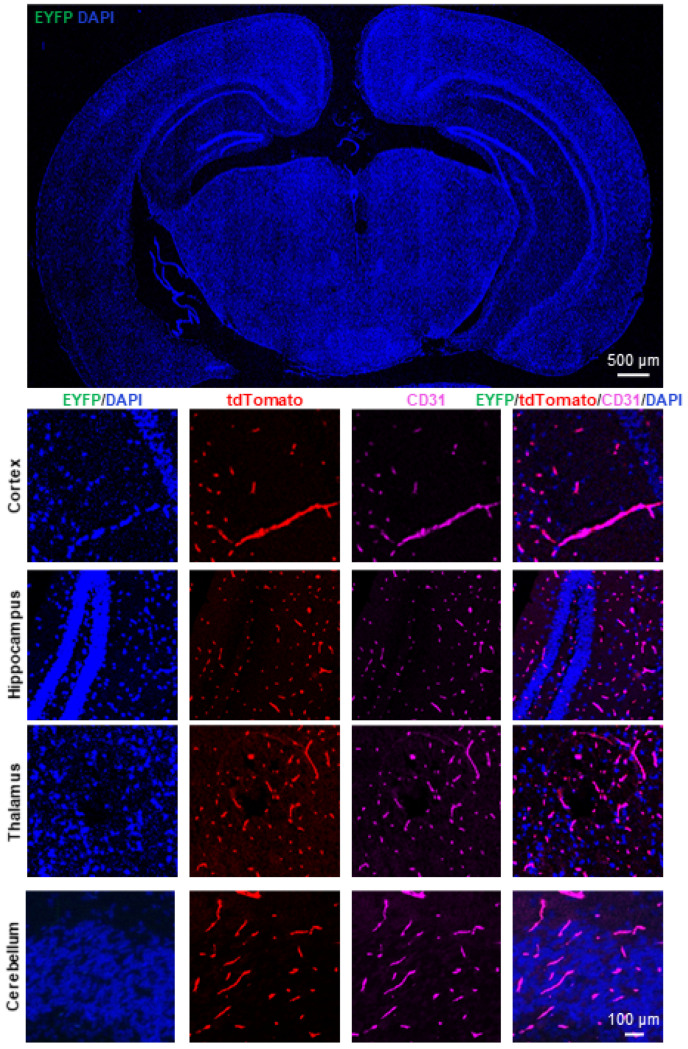
Examination of the leakiness of activity in *Slco1a4-Cre^ERT2^* and *ROSA26^LSL-EYFP^* mice. EYFP (green), tdTomato (red), and CD31 (purple) immunostaining analysis in the brain of *Slco1a4-Cre^ERT2^* and *ROSA26^LSL-EYFP^* mice in the absence of TAM. No EYFP^+^ cells (green) were found in the brain in *Slco1a4-Cre^ERT2^* and *ROSA26^LSL-EYFP^* mice without TAM.

## Data Availability

All data supporting the findings in this study are available upon request from the corresponding author.

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
