# Peer review of "Generation of Slco1a4-CreERT2-tdTomato Knock-in Mice for Specific Cerebrovascular Endothelial Cell Targeting"

_ijms, 2024, doi:10.3390/ijms25094666_

Round 1

Reviewer 1 Report

Comments and Suggestions for Authors

The authors of the manuscript present results from an original experimental study in which they search for a marker, associated specifically with the physiological functions and/or signaling pathways of the cerebrovascular endothelial cells. They have used a publicly available single-cell RNAseq database, which was screened for a solute carrier organic anion transporter family, member 1a4 (Slco1a4) as a candidate marker of cerebrovascular endothelial cells. According to the author's results such a marker was the endogenous Slco1a4, expressed in almost all cerebrovascular endothelial, but not in any other non-endothelial cell types in the brain. 

The study and the results, presented in the manuscript are topical and important from scientific, and clinical point of view. Cerebrovascular diseases (mostly the atherosclerotic and cardioembolic ones), together with the cardiovascular diseases are leading causes for morbidity and premature death worldwide, particularly in the industrially developed countries.  While the diagnostic and therapeutic strategies for cardiac diseases have evolved a lot during the last 20 years with many new markers for cardiac damage and new groups of drugs implemented in the clinical practice, diagnostic laboratory markers and pharmacological options for treatment of atherosclerotic cerebrovascular disease are still relatively limited. From this point of view, any new research related to evaluation of cerebrovascular physiological functions/dysfunctions and/or signaling pathways  is applausable. 

I have the following recommendations to the authors of the manuscript:

1. The title must be modified, so that it becomes more attractive to readers (particularly clinicians).

2. "Materials and methods" are expected to be positioned before "Results" and "Discussion".

3. At the end of "Discussion" there should be "Study limitations".

Reviewer 2 Report

Comments and Suggestions for Authors

This is a very nice paper describing a new brain endothelial specific CreERT2 recombinase for use in mouse genetic studies.  The authors used publically available data to identify a gene/transcript that is only expressed in brain endothelial cells. They then inserted CreERT2 along with a tdTomato lineage tracer into the Slco1a4 locus using Cre recombination.   The authors characterize the expression pattern to show that the expression is brain endothelial specific.

This tool should become very useful to the mouse genetics community who are studying brain vascular diseases.  

I have only a few suggestions.  

1. In Figure 4, the tdTomato appears to at least partially overlap with the PDGFRbeta (pericytes) and alpha SMA (smooth muscle cells).    Since these are the most important cell types associated with the vasculature that are not endothelial cells, they need to show much better that the tdTomato is not in the same cells as PDGFRbeta and alpha SMC.  One way to do this would be single cell RNA sequencing of some brain tissue.  Or a much higher magnification view of the immunofluorescence to show that they do not overlap.  

2. In the Discussion the authors should discuss another brain specific CreERT2 that is being used by a handful of people.   Slco1c1(BAC)-CreERT2.   The Cre is briefly described in

Ridder DA, Lang MF, Salinin S, Roderer JP, Struss M, Maser-Gluth C et al. TAK1 in brain endothelial cells mediates fever and lethargy. The Journal of experimental medicine. 2011;208(13):2615–23. doi: 10.1084/jem.20110398.

This Slco1c1 Cre has been successfully used to delete disease genes to create a brain specific cerebrovascular phenotype.

Detter MR, Shenkar R, Benavides CR, Neilson CA, Moore T, Lightle R, Hobson N, Shen L, Cao Y, Girard R, Zhang D, Griffin E, Gallione CJ, Awad IA, Marchuk DA.  Novel Murine Models of Cerebral Cavernous Malformations.  Angiogenesis 23:651-66, 2020.  

Ren AA, Snellings DA, Su YS, Hong CC, Castro M, Tang AT, Detter MR, Hobson N, Girard R, Romanos S, Lightle R, Moore T, Shenkar R, Benavides C, Beaman MM, Müller-Fielitz H, Chen M, Mericko P, Yang J, Sung DC, Lawton MT, Ruppert JM, Schwaninger M, Körbelin J, Potente M, Awad IA, Marchuk DA, Kahn ML. PIK3CA and CCM mutations fuel cavernomas through a cancer-like mechanism. Nature 594:271-276, 2021. 

Round 2

Reviewer 2 Report

Comments and Suggestions for Authors

The authors have responded to my concerns